# Influence of Hydrophobic Side-Chain Length in Amphiphilic Gradient Copoly(2-oxazoline)s on the Therapeutics Loading, Stability, Cellular Uptake and Pharmacokinetics of Nano-Formulation with Curcumin

**DOI:** 10.3390/pharmaceutics14122576

**Published:** 2022-11-23

**Authors:** Shubhashis Datta, Veronika Huntošová, Annamária Jutková, Róbert Seliga, Juraj Kronek, Adriána Tomkova, Lenka Lenkavská, Mariana Máčajová, Boris Bilčík, Barbora Kundeková, Ivan Čavarga, Ewa Pavlova, Miroslav Šlouf, Pavol Miškovský, Daniel Jancura

**Affiliations:** 1Center for Interdisciplinary Biosciences, Technology and Innovation Park, P. J. Safarik University in Košice, Jesenna 5, 04154 Košice, Slovakia; 2Department of Biophysics, Faculty of Science, P. J. Safarik University in Košice, Jesenna 5, 04154 Košice, Slovakia; 3SAFTRA Photonics s.r.o., Moldavska Cesta 51, 04011 Košice, Slovakia; 4Department for Biomaterials Research, Polymer Institute of the Slovak Academy of Sciences, Dubravska Cesta 9, 845 41 Bratislava, Slovakia; 5Institute of Animal Biochemistry and Genetics, Centre of Biosciences, Slovak Academy of Sciences, Dubravska Cesta 9, 840 05 Bratislava, Slovakia; 6Institute of Macromolecular Chemistry, Czech Academy of Sciences, Heyrovskeho Nam. 2, 162 06 Prague, Czech Republic; 7Cassovia New Industry Cluster, Tr. SNP 1, 04001 Košice, Slovakia

**Keywords:** polymeric nanoparticles, hydrophobicity, curcumin, cancer cells, structure-property–relationship

## Abstract

Due to the simple one-step preparation method and a promising application in biomedical research, amphiphilic gradient copoly(2-oxazoline)s are gaining more and more interest compared to their analogous block copolymers. In this work, the curcumin solubilization ability was tested for a series of amphiphilic gradient copoly(2-oxazoline)s with different lengths of hydrophobic side-chains, consisting of 2-ethyl-2-oxazoline as a hydrophilic monomer and 2-(4-alkyloxyphenyl)-2-oxazoline as a hydrophobic monomer. It is shown that the length of the hydrophobic side-chain in the copolymers plays a crucial role in the loading of curcumin onto the self-assembled nanoparticles. The kinetic stability of self-assembled nanoparticles studied using FRET shows a link between their integrity and cellular uptake in human glioblastoma cells. The present study demonstrates how minor changes in the molecular structure of gradient copoly(2-oxazoline)s can lead to significant differences in the loading, stability, cytotoxicity, cellular uptake, and pharmacokinetics of nano-formulations containing curcumin. The obtained results on the behavior of the complex of gradient copoly(2-oxazoline)s and curcumin may contribute to the development of effective next-generation polymeric nanostructures for biomedical applications.

## 1. Introduction

Nanoparticles formed by the controlled aggregation of amphiphilic block or gradient copolymers are one of the most promising delivery systems that improve the accumulation of poorly water-soluble anticancer drugs or therapeutics in a tumor and reduce their side effects [1,2,3,4,5,6,7,8,9]. Block and gradient copolymers are two types of macromolecules that differ in the distribution of monomers. Block copolymers exhibit an abrupt change in monomer distribution, while gradient or pseudo-diblock copolymers exhibit a gradual change in monomer distribution along the polymer chain [10,11]. Similar to diblock copolymers, the aggregation behavior of gradient copolymers can be triggered by pH [12], temperature [13], or solvent composition [14]. The better solubility of gradient copolymers in a solution compared to the block copolymers with the same composition leads to a higher critical aggregation concentration [15]. However, gradient copolymers possess a lower-value aggregation number, as evidenced by the smaller size of the formed nanoparticles, as detected by dynamic light scattering [16].

In the last decade, poly(2-oxaziline)s (POx) have been widely used by researchers as an alternative to poly (ethylene glycol) [17,18,19], showing their potential use as biomaterials in medical applications and thus being considered as next-generation polymer therapeutics. The main application of POx is in the preparation of drug conjugates, micelles, and hydrogels and drug and DNA chain delivery [20,21,22,23,24]. A variety of POx structures with variable chemical and physical properties are synthesized by living cationic ring-opening polymerization (CROP) of 2-oxazolines [25,26]. The gradient copoly(2-oxazoline)s exhibited some interesting structural phenomena and formed more compact and smaller particles compared to the analogous block copolymers [27]. Numerous research papers have reported the promising potential of amphiphilic block copoly(2-oxazoline)-based delivery systems for the entrapment of anticancer drugs with an exceptionally high loading capacity and targeted delivery to cancer cells [28,29,30]. Recently, attempts have been made to investigate the potential of amphiphilic gradient copoly(2-oxazoline)s for the formulation of hydrophobic drugs [31,32,33,34]. Poly(2-methyl-2-oxazoline-*grad*-2-phenyl-2-oxazoline) (PMeOx-*grad*-PhOx) is the most studied gradient copoly(2-oxazoline) for drug encapsulation, containing 2-methyl-2-oxazoline (MeOx) as a hydrophilic monomer and 2-phenyl-2-oxazoline (PhOx) as a lipophilic monomer. Hruby and co-workers reported that gradient copoly(2-oxazoline)s exhibit significantly higher loading capacity and entrapment efficiency than their analogous block copolymers at certain comonomer ratios. The gradient copolymers contained 2-phenyl-2-oxazoline (PhOx), 2-(4-butylphenyl)-2-oxazoline (BuPhOx), or 2-(4-butoxyphenyl)-2-oxazoline (BuOPhOx) as the hydrophobic monomer [33]. Recently, we synthesized a small library of gradient copoly(2-oxazoline)s with different hydrophobic side-chain lengths containing 2-ethyl-2-oxazoline (EtOx) as a hydrophilic monomer and 2-(4-methoxyphenyl)-2-oxazoline (MeOPhOx), 2-(4-ethoxyphenyl)-2-oxazoline (EtOPhOx), 2-(4-hexyloxyphenyl)-2-oxazoline (HexOPhOx), or 2-(4-dodecyloxyphenyl)-2-oxazoline (DOPhOx) as a hydrophobic monomer [35,36]. The self-assembly of these gradient copoly(2-oxazoline)s has been used to formulate hypericin, a promising compound for photodiagnostics and photodynamic therapy [35]. The hypericin’s loading capacity and entrapment efficiency did not change significantly with the increase in the length of the hydrophobic side-chain in the synthesized gradient copoly(2-oxazoline)s [35]. However, the in vitro and in vivo release of hypericin from the nanoparticles and delivery to the cancer cells were strongly influenced by the length of the hydrophobic alkyl chain of the copolymers [35].

Despite recent advances in amphiphilic gradient copoly(2-oxazoline)s, all studies have been limited to self-aggregation and drug encapsulation. Therefore, there is a lack of studies on the effects of structural changes in amphiphilic gradient copoly(2-oxazoline)s on therapeutic drug loading, stability, and activity of nanoformulation, which will be addressed in this work. We formulated curcumin by self-assembling a series of gradient copoly(2-oxazoline)s synthesized in our previous work [35] to investigate the role of increasing hydrophobic side-chain length on the properties of the complexes of the drug (curcumin) and polyoxazoline nanoparticles.

Curcumin is a naturally occurring, bioactive, polyphenol-derived compound with diverse biological properties, including anticancer, antioxidant, antiamyloid, antibiotic, and antiviral properties [37,38,39,40]. However, it has recently been classified as both PAINS (pan-assay interference compounds) and IMPS (invalid metabolic panaceas) [39]. Similar to other hydrophobic drugs, curcumin is poorly soluble in water at neutral pH (logP ranges from 2.3 to 3.2) and room temperature [41]. Several attempts have been made to increase the solubility of curcumin and deliver it to cancer cells using the self-assembly of block copoly(2-oxazoline)s [42,43,44,45]. Recently, we demonstrated the potential of one of the copolymers from the library of amphiphilic gradient copoly(2-oxazoline)s we synthesized for the encapsulation of curcumin [36]. The nanoparticles (NPs) loaded with curcumin showed excellent in vitro stability and low toxicity.

Specifically, in this work we investigated the kinetic stability of the self-assembled nanostructure under physiological conditions to mimic the cellular/tissular environment. In addition, the cellular uptake and cytotoxicity of curcumin-loaded Pox-NPs in cancer cells U87 MG (human gliobastoma cells) were investigated by confocal fluorescence imaging and flow cytometric measurements, and the biodistribution of curcumin-loaded NPs was evaluated by assay on the chorioallantoic membrane (CAM) of avian embryos. The present study highlights the important interplay between the molecular structure, therapeutics loading, stability, and cellular uptake of gradient Pox-NPs, which may be helpful in developing effective next-generation polymeric nanostructures for biomedical applications.

## 2. Materials and Methods

### 2.1. Materials

Curcumin (MW = 368.4, ≥94% purity from *Curcuma longa*), dimethylsulfoxide (DMSO, anhydrous, ≥99.9%), human serum albumin (agarose gel electrophoresis, purity ≥96%), γ-globulins from bovine blood (agarose gel electrophoresis, purity ≥99%), 3,3′-dioctadecyloxacarbocyanine perchlorate (DiO, ≥98% TLC), and 1,1′-dioctadecyl- 3,3,3′,3′-tetramethylindocarbocyanine perchlorate (DiI, ≥97% TLC) were purchased from Sigma Aldrich (Taufkirchen, Germany). Fetal bovine serum albumin (FBS) was purchased from Biosera, Nuaille, France, and the dialysis membrane (3.5 kDa MWCO) and syringe filter (0.22, or 0.45 µm pore size) were purchased from Merck-Millipore (Darmstadt, Germany).

### 2.2. Preparation of Curcumin-Loaded POx NPs

The synthesis of gradient copolymers was reported in our previous study [35,36], and (EtOx)_88_-*grad*-(MeOPhOx)_12,_ (EtOx)_88_-*grad*-(EtOPhOx)_12_, (EtOx)_88_-*grad*-(HexOPhOx)_12_, and (EtOx)_88_-*grad*-(DOPhOx)_12_ with varying hydrophobic side-chain lengths and a fixed theoretical molar fraction ratio (88:12) between hydrophilic and hydrophobic units were used for the formulation of curcumin. Curcumin-loaded polymeric NPs were prepared by the nanoprecipitation method as reported in our previous study [35,36]. The copolymer and curcumin were dissolved in DMSO with constant stirring at 600 rpm for 10 min at room temperature. The feeding concentration of polymer was 4 mg/mL and the feeding concentration of curcumin was 0.036 mg/mL or 1 mg/mL. An equal volume of phosphate-buffered saline (PBS, Sigma-Aldrich Germany) of pH 7.4 and distilled water was added dropwise with constant stirring at 600 rpm for another 10 min. The solution mixture was dialyzed (MWCO 3.5 kDa) against PBS for 36 h. PBS was changed every 2 h for the first 6–8 h. 

### 2.3. Dynamic Light Scattering (DLS)

The size of NPs was measured on a Zetasizer Nano ZS (Malvern Instrument, Malvern, UK) using a He/Ne-laser (λ_ex_ = 633 nm) at a scattering angle of 173°. The autocorrelation functions of the scattered intensity were analyzed by means of the Cumulants method to yield the hydrodynamic diameters. The concentration of curcumin-loaded POx NPs was kept at 600 µg/mL and they were filtered through a 0.45 μm syringe filter before measurement. 

### 2.4. Calculation of Curcumin Loading Capacity and Encapsulation Efficiency

After 36 h of dialysis, the NPs solutions were filtered through a 0.22 or 0.45 μm syringe filter to remove the precipitated drug, which would provide false efficiency. Then the solutions were freeze-dried, and dried samples were dissolved in DMSO. UV-Vis absorption spectroscopy was used to calculate the concentration of curcumin inside NPs. Since curcumin was found to form an inclusion complex with polymers, the calibration curve in DMSO was formulated using different concentrations of curcumin (0.003 mg/mL to 0.01 mg/mL) in the presence of the respective gradient polymers. The loading capacity (wt%) and encapsulation efficiency (%) were determined using the procedure described in our previous study [35]. 

### 2.5. Transmission Electron Microscopy (TEM) and Cryogenic Transmission Electron Microscopy (Cryo–TEM)

Transmission electron microscopy (TEM) and cryogenic transmission electron microscopy (cryo–TEM) observations of curcumin-loaded polymeric NPs were performed using a Tecnai G2 Spirit Twin 12 (FEI, Prague, Czech Republic), equipped with cryo-attachment (Gatan, cryo-specimen holder) using a bright field imaging mode at an accelerating voltage of 120 kV. In order to minimize the misinterpretation of the results due to possible sample preparation artifacts, three independent visualization techniques were applied to each sample: Imaging of dried nanoparticles deposited onto a thin carbon film after fast removal of the solution (TEM after fast solution removal), imaging of the dried nanoparticles negatively stained with uranyl acetate (TEM after negative staining), and imaging of thin frozen films containing the NPs using cryogenic microscopy (cryo-TEM).

Regarding the first method (TEM after fast solution removal), the sample solutions (concentration of polymer 4 mg/mL) were dropped onto a copper TEM grid (300 mesh) coated with a thin, electron-transparent carbon film. After 1 min, the solution was removed by touching the grid bottom with filtering paper in order to minimize oversaturation during the drying process. Before TEM observation, the specimens were left to dry completely at room temperature. 

The second method (TEM after negative staining) consisted of three steps: (i) Hydrophilization of the grids covered by a thin carbon film by glow discharge using an Expanded Plasma Cleaner (Harrick Plasma, Ithaca, NY, USA), (ii) deposition of the sample onto the hydrophilized carbon film followed by removal of the solution in exactly the same way as in the previous method, and (iii) negative staining with a uranyl acetate solution (2 wt%) as described elsewhere [46]. It is worth noting that the hydrophilization step was necessary so that the uranyl acetate was deposited regularly onto the grid. 

The last method (cryo-TEM) employed a completely different sample preparation: 3 μL of the sample solution was deposited on an electron microscopy grid covered with a lacey carbon supporting film (Agar Scientific, Essex, UK) after hydrophilization by glow discharge (glow discharge performed as in the previous method). The solution excess was removed by blotting (Whatman no. 1 filter paper) for ∼1 s and then the grid was immediately plunged into liquid ethane held at −181 °C. The frozen sample was transferred to the microscope and observed at −173 °C under the conditions described above (120 kV, bright field imaging). 

### 2.6. UV-Vis Absorbance and Steady-State Fluorescence Spectroscopy

The UV-Vis absorption spectra were recorded using a Shimadzu UV-2401 (Kyoto, Japan) spectrophotometer and the steady-state fluorescence emission spectra were recorded using a spectrofluorimeter Shimadzu RF-5301 (Kyoto, Japan). 

We prepared DiO and DiI co-loaded FRET-active polymeric NPs and investigated their structural integrity in the presence of fetal bovine serum (FBS), human serum albumin (HSA), and γ-globulin solutions in PBS of pH 7.4. 

Four different DiO and DiI co-loaded FRET polymeric NPs, (EtOx)-*grad*-(MeOPhOx), (EtOx)-*grad*-(EtOPhOx), (EtOx)-*grad*-(HexOPhOx), and (EtOx)-*grad*-(DOPhOx) NPs, were prepared by dissolving polymer (4 mg/mL), DiO (0.3 wt%) and DiI (0.3 wt%) in DMSO and following the procedure mentioned in Section 2.2. FRET-active polymeric NPs with a final concentration of 800 µg/mL were incubated with an FBS (10%, *v*/*v*) solution in PBS, or HSA (35 mg/mL) or γ-globulin (15 mg/mL) solutions in PBS of pH 7.4 with gentle agitation. The volume ratio of nanoparticle solutions to those solutions was 1:1. Fluorescence spectra of those solutions were recorded over 24 h with an excitation wavelength of 484 nm.

### 2.7. Cell Culture

The cancer cells of U87 MG (human glioblastoma cells, Cells Lines Services, Eppelheim, Germany) lines were grown in the cell culture medium (high glucose, GlutaMAX^TM^ supplement, pyruvate Dulbecco’s modified Eagle medium (D-MEM, Gibco-Invitrogen, Life Technologies Ltd., Paisley, UK) at 80% confluence in the dark at 37 °C in a 5% CO_2_ and humidified atmosphere. The complete cell culture medium was prepared with 10% (*v*/*v*) FBS. 

### 2.8. Confocal Fluorescence Imaging

The curcumin fluorescence in U87 MG cells was detected 1 h after its administration alone and in the presence of polymeric nanoparticles (λ_exc_ = 405 nm, λ_em_ = 490–540 nm). The distribution of curcumin in cells was observed with the inverted confocal microscope (LSM700, Zeiss, Oberkochen, Germany), equipped with the oil immersion objective (63X, NA = 1.46, with adjustable coverslip correction) and a CCD camera (AxioCam HRm, Zeiss, Germany). The obtained fluorescence images were analyzed in Zen 2011 software (Zeiss, Germany) or ImageJ software (National Institutes of Health, Bethesda, MD, USA). The endoplasmic reticulum was stained with 1 µM Invitrogen™ ER-Tracker™ Green (BODIPY™ FL Glibenclamide) (ThermoFisher Scientific, Waltham, MA, USA) for live-cell imaging for 30 min (excitation at 488 nm and emission in the spectral range 490–530 nm). The nuclei were stained with 10 µg/mL Hoechst 33258 (ThermoFisher Scientific, Waltham, MA, USA) for 30 min (excitation at 405 nm and emission in the spectral range 450 ± 40 nm). The plasma membrane of the cells was stained with 2.5 μg/mL CellMask^TM^ Orange (ThermoFisher Scientific, Waltham, MA, USA) for 5 min (excitation at 555 nm and emission in the spectral range > 580 nm).

### 2.9. Flow Cytometry

The curcumin uptake by U87 MG cells was analyzed with a flow cytometer (MACSQuant^®^ Analyzer, Miltenyi, Bergisch Gladbach, Germany) in the B1 channel at excitation λ_exc_ = 488 nm and emission λ_em_ = 525/50 nm. The 10 µM curcumin alone and loaded in polymeric nanoparticles was detected 0.1, 1, 3, 5, and 24 h after administration at 37 °C. The cells were washed with ice-cold PBS, detached with trypsin/EDTA (Gibco-Invitrogen, Life Technologies Ltd., Paisley, UK), centrifuged at 1200 rpm for 10 min, and the pellets were resuspended in ice-cold PBS.

The active transport was studied in U87 MG cells detached with trypsin/EDTA, centrifuged, and resuspended in D-MEM without serum components. The curcumin alone and curcumin-loaded polymeric nanoparticles aliquots were administered to 0.5 mL solutions with cell suspension and maintained at 37 and 4 °C for 1 h in small Eppendorf tubes before curcumin fluorescence detection.

### 2.10. MTT-Assay

The metabolic activity (relative to cytotoxicity) of U87 MG cells in the presence of 10^−7^–10^−3^ M curcumin and curcumin-loaded polymeric nanoparticles was assessed by MTT (3-(4,5-dimethylthiazol-2-yl)-2,5-diphenyltetrazolium bromide, Sigma-Aldrich, Taufkirchen, Germany) assay. The purple formazan was detected at 560 nm and 750 nm by a 96-well plate absorption reader (GloMax^®^-Multi+ Detection System with Instinct Software, Promega Corporation, Fitchburg, WI, USA). The cells were treated for 24 and 48 h with curcumin and curcumin-loaded polymeric nanoparticles in dark conditions. An MTT assay was performed according to the supplier’s protocol. The error bars represent standard deviations from the experimental data mean values (8 values, repeated at least in triplicate). The level of significance was estimated with the Student’s *t*-test: * *p* < 0.05, ** *p* < 0.01, *** *p* < 0.001.

### 2.11. CAM Model Preparation

The chorioallantoic membrane (CAM) of fertilized Japanese quail (*Coturnix japonica*) eggs (a breeding colony of IABG SAS) was used as a pre-clinical model system. The eggs were incubated in a forced draught incubator (MIDI BIOS, Sedlcany, Czech Republic) at 37.5 °C and 50–60% relative humidity. The ex ovo cultures were prepared as follows: The surface of the eggs was disinfected with a 70% ethanolic solution at embryonic day 3 (ED3) in a sterile laminar flow hood. Subsequently, the eggs were opened with scissors, and the embryos were transferred into the six-well tissue culture plates (Sarstedt, Germany) and kept in a 37 °C humidified incubator (Memmert, Buchenbach, Germany) until ED11.

### 2.12. Curcumin Fluorescence Biodistribution in CAM

The curcumin in PBS at pH = 7.4 and in the polymeric nanoparticles was topically applied to the CAM via the silicone rings as 30 µL aliquots on ED9. The concentration of curcumin was maintained at 10 µM in the 30 µL solutions. The fluorescence of curcumin in CAM tissue was recorded with a digital camera (Canon EOS 6D II with Canon MP-E 65 mm f/2.8, macro lens, Canon, Tokyo, Japan) in intervals 0, 1, 3, 5, and 24 h after compound administration. The CAMs were illuminated using either white light (ring flash Canon MR-14EX, Tokyo, Japan) or custom-made circular LED light emitted at 405 nm. The fluorescence intensity was evaluated using Image J software (NIH, Bethesda, MA, USA). The red intensity of RGB images was normalized to the maximum values. Whole-image profile plots (ImageJ plugin) were derived each time after curcumin administration: Before administration and from 0 (right after administration) up to 24 h.

### 2.13. CAM Tissue Histology

The histological sections of the CAM tissue were analyzed 24 h after curcumin and curcumin-loaded polymeric nanoparticles administration. CAM tissue was separated and fixed with 4% paraformaldehyde (Sigma-Aldrich, Germany). Subsequently, 5 µm paraffin sections were prepared for histopathological analysis to determine the photodamage of the tissue (hematoxylin (BAMED, Malacky, Slovakia) and eosin (BAMED, Slovakia) staining). The sections were evaluated with a light microscope Kapa 2000 (Kvant, Bratislava, Slovakia), 10× and 20× objective, and a digital camera (Canon EOS 6D II, Canon, Tokyo, Japan).

## 3. Results and Discussion

### 3.1. Effect of Hydrophobic Side-Chain Length on Size, Morphology, and Therapeutics Loading

Four different gradient copoly(2-oxazoline)s (Figure 1 and Table 1) with different hydrophobic side-chain lengths, synthesized in our previous studies [35,36], were used here for the encapsulation of curcumin. Curcumin-loaded NPs were prepared via self-assembly of these gradient copolymers (Figure 1) using the nanoprecipitation method [35,36]. The feeding concentrations of copolymers and curcumin were set at 4 mg/mL and 0.036 mg/mL, respectively. The size of curcumin-loaded gradient POx NPs was measured by DLS (Appendix A) after 24 h and 96 h from the time of their preparation. The volume-weighted average size (*D_h_*) after 24 h is summarized in Table 2.

The volume-weighted average size of four curcumin-loaded gradient POx NPs, (EtOx)_88_-*grad*-(MeOPhOx)_12_, (EtOx)_88_-*grad*-(EtOPhOx)_12_, (EtOx)_88_-*grad*-(HexOPhOx)_12_, and (EtOx)_88_-*grad*-(DOPhOx)_12_ NPs was <100 nm (Table 1), which is suitable for targeting cancer tissue due to the enhanced permeation and retention (EPR) effect [47]. On the other hand, the intensity-weighted size distribution plot (Appendix A) after 24 h also showed the presence of larger NPs, which accounted for only a small volume fraction of the particles in the solutions. A size below 100 nm indicates the absence of larger aggregates found in the volume-weighted size-distribution plot of unloaded gradient POx NPs [35]. In the present study, the presence of a small amount of curcumin (the curcumin concentration in the feed was 0.036 mg/mL, ~1 wt%) resulted in the formation of small and well-defined particles. As the length of the hydrophobic side-chain in the gradient copolymers increased, the volume-weighted average size of curcumin-loaded POx NPs increased slightly after 24 h (Table 2). The volume-weighted average size of (EtOx)_88_-*grad*-(MeOPhOx)_12_ NPs was 21 nm (PDI = 0.3), in (EtOx)_88_-*grad*-(EtOPhOx)_12_ NPs it was 24 nm (PDI = 0.35), in (EtOx)_88_-*grad*-(HexOPhOx)_12_ NPs it was 36 nm (PDI = 0.27), and in (EtOx)_88_-*grad*-(DOPhOx)_12_ NPs it was 45 nm (PDI = 0.18). Hruby and co-workers investigated the effect of hydrophobicity in a series of amphiphilic gradient copoly(2-oxazoline)s on the size of NPs loaded with the antibiotic rifampicin [33]. In all cases, 2-methyl-2-oxazoline (MeOx) was used as the hydrophilic monomer, while the hydrophobic monomer differed in each series: 2-phenyl-2-oxazoline (PhOx), 2-(4-butphenyl)-2-oxazoline (BuPhOx), or 2-(4-butoxyphenyl)-2-oxazoline (BuOPhOx). In addition, the copolymers varied in the ratio of hydrophilic and hydrophobic units from 90:10, 80:20, or 70:30. Interestingly, increasing hydrophobicity in gradient copolymers with a ratio of 90:10 did not affect the size of rifampicin-loaded NPs formed by their self-assembly. This is a very similar size variation to the gradient copolymers we synthesized, with the exception of (EtOx)_88_-*grad*-(DOPhOx)_12_ with almost the same mixing ratio (88:12). However, Hruby and co-workers observed a strong dependence of hydrophobicity on the size of NPs at two other compositional ratios: 80:20 and 70:30. 

The morphology of curcumin-loaded POx NPs was studied by TEM microscopy. TEM microscopy showed that the particles appeared as elongated worm-shaped NPs. A representative sample of TEM microscopy results with curcumin-loaded (EtOx)_88_-*grad*-(HexOPhOx)_12_ NPs is shown in Figure 1. Standard TEM micrographs (bright-field image of POx NPs deposited on a carbon film after the rapid removal of the solution) did not show clear structures due to low contrast, but the other two TEM techniques (TEM after negative staining and Cryo-TEM) confirmed that nanoparticles were worm-shaped particles with a diameter of approximately 40 nm and a variable length from 20 nm to 300 nm. It is worth noting that the same morphology of worm-shaped NPs was observed using two completely independent sample preparation techniques: The rapid removal of solution (Figure 1a) and rapid freezing of the solution (Figure 1b). Although the negatively stained worm-shaped NPs look thicker than the same NPs visualized by cryogenic microscopy, the measurement in ImageJ [48] revealed that the average diameter of the bright fibers in Figure 1a was exactly the same as the diameter of the dark fibers in Figure 1b, i.e., the apparent greater thickness of the negatively stained particles is due to the staining agent surrounding the NPs.

The worm-shaped NPs were observed for all systems studied, regardless of the length of the hydrophobic side-chain. However, TEM microscopy of gradient copolymers with short side-chain lengths, (EtOx)_88_-*grad*-(MeOPhOx)_12_ and (EtOx)_88_-*grad*-(EtOPhOx)_12_, revealed the presence of both spherical and worm-shaped structures, with the population of spherical NPs being comparatively low. On the other hand, cryo SEM microscopy of curcumin-loaded (EtOx)_88_-*grad*-(DOPhOx)_12_ copolymer NPs also confirmed the presence of spherical particles [36]. Kabanov and co-workers observed that poly(2-oxazoline)s-based block copolymer micelles can be elongated over time, which is confirmed by DLS and TEM measurements [49]. In our case, the DLS result after 96 h (Appendix A) indicates an increase in the population of elongated structures of curcumin-loaded NPs. TEM measurements were performed after 2 weeks from the time of NPs production. Thus, it is possible that the concentration of worm-shaped NPs increased during the shelf-life of our sample. Kabanov and co-workers also observed that such morphological transitions strongly depend on the type of drug loaded into the micelles, and based on this fact, they divided the drugs into two categories: “worm-permissive” and “worm-inhibiting” [49]. These two categories of drugs showed great differences in drug–polymer interactions. The gradient copolymers synthesized in our case, (EtOx)_88_-*grad*-(MeOPhOx)_12_, (EtOx)_88_-*grad*-(EtOPhOx)_12_, and (EtOx)_88_-*grad*-(HexOPhOx)_12_, self-assembled in the presence of hypericin, showed (TEM microscopy) spherical morphology [35]. Thus, we can assume that curcumin and hypericin interact with the core of gradient copoly(2-oxazoline)s in different ways, which is a plausible reason for the two different morphologies. 

The curcumin-loading capacity and encapsulation efficiency of self-assembled nanostructures of four gradient copolymers, (EtOx)_88_-*grad*-(MeOPhOx)_12_, (EtOx)_88_-*grad*-(EtOPhOx)_12_, (EtOx)_88_-*grad*-(HexOPhOx)_12_, and (EtOx)_88_-*grad*-(DOPhOx)_12_, were determined using the equations described in our previous study [35]. Here, two different concentrations of curcumin in feed were used: 0.036 mg/mL and 1 mg/mL. The curcumin-loading capacity of (EtOx)_88_-*grad*-(DOPhOx)_12_ NPs reported in a previous study [36] using 0.036 mg/mL curcumin in the feed was recalculated here. Figure 2A,B show the variation in the curcumin-loading capacity as a function of the hydrophobic side-chain length. The solubilization of curcumin in the presence of gradient copolymers increased as the length of the hydrophobic side-chain (R) in (ROPhOx) increased from methyl to hexyl, independent of the concentration of curcumin in the feed. However, a further increase in the side-chain length in (EtOx)_88_-*grad*-(DOPhOx)_12_ resulted in a decrease in the curcumin-loading ability. Thus, the large dodecyl group did not function better than the hexyl group. This is likely due to the crystallization induced by this larger alkyl group, which does not allow interaction with the drug, and due to the steric effect of the side chain. However, the crystallinity within the aggregates could not be confirmed by experimental methods. This is likely due to the crystallization induced by this larger alkyl group that does not allow interaction with the drug. In the present study, a maximum curcumin loading of 15.3 wt% was achieved with the (EtOx)_88_-*grad*-(HexOPhOx)_12_ copolymer when the curcumin concentration in the feed was 1 mg/mL. When the curcumin concentration in the feed increased from 0.036 mg/mL to 1 mg/mL, the encapsulation efficiency of the gradient copolymers decreased significantly, except in the case of (EtOx)_88_-*grad*-(HexOPhOx)_12_ (Figure 2C).

Hoogenboom and co-workers investigated the influence of the chain length of analogous gradient and block copoly(2-oxazoline)s on curcumin loading, observing a strong dependence of self-assembly properties on the total length of the polymer [50]. Using gradient copoly(2-oxazoline)s containing 2-methyl-2-oxazoline (MeOx) as a hydrophilic monomer and 2-phenyl-2-oxazoline (PhOx) as a hydrophobic monomer, they achieved curcumin loading of approximately 16 wt%. They also showed that drug loading in the preparation of NPs by the direct dissolution method depends on the hydrophobic content of the copolymers. The advantages of using PhOx as a hydrophobic monomer are that it favors the encapsulation of hydrophobic natural products and drugs with the π-system, such as curcumin, via the π–π stacking interaction [43] and its rigid “glassy” core prevents the premature release of drugs from NPs [51]. Gradient copolymers with different chain lengths (R) in the (ROPhOx) part of the copolymers were used for the formulation of hypericin [35]. The results showed a minimal effect of the hydrophobic side-chain length on hypericin loading. However, in the present study, the hydrophobic side-chain length of the synthesized gradient copoly(2-oxazoline)s plays a crucial role in curcumin loading, indicating the need for the optimal side-chain length to achieve maximum therapeutic loading.

### 3.2. Effect of Hydrophobic Side-Chain Length on the Stability of Nano-Formulations

The structural integrity of polymeric NPs after intravenous administration is a prerequisite for the delivery of encapsulated anticancer drugs. Once a nanoparticle loses its integrity in the blood, it immediately releases the transported drug into the bloodstream [52,53]. NPs encounter several bimolecular species in the blood that can interact with them and affect their structure. Serum proteins, i.e., albumin, lipoproteins, and globulins, are the main protein components of blood plasma responsible for NPs disassembly and premature drug leakage [54,55]. The kinetic stability of NPs describes their behavior over time in an aqueous medium. FRET has been extensively used to monitor the structural integrity of polymeric NPs in vitro and in vivo [56,57,58,59,60]. Fluorophores such as 3,3′^¢^-dioctadecyloxacarbocyanine perchlorate (DiO) as a donor and 1,1′-dioctadecyl-3,3,3′,3′-tetramethylindocarbocyanine perchlorate (DiI) as an acceptor are commonly used in FRET to evaluate the kinetic stability of empty or drug-encapsulated NPs in biological medium. The appearance of FRET is sensitive to the variation of the distances between the donor and acceptor molecules and is only effective when the donor and acceptor molecules are close enough (2–10 nm) [57]. When both FRET-active molecules (DiO as the donor and DiI as the acceptor) are encapsulated in POx NPs and excited at the appropriate wavelength, energy transfer occurs between these two fluorophore molecules due to their spatial proximity. However, when the POx NPs disassemble or lose their integrity, the FRET-active molecules are released and diffuse, disrupting the energy transfer. The objective of the present study was to investigate the influence of the length of the hydrophobic side-chain in gradient copoly(2-oxazoline)s on the structural integrity of their self-assembled NPs under physiological conditions to mimic a cellular/tissular environment. For this purpose, four different NPs, (EtOx)_88_-*grad*-(MeOPhOx)_12_, (EtOx)_88_-*grad*-(EtOPhOx)_12_, (EtOx)_88_-*grad*-(HexOPhOx)_12_, and (EtOx)_88_-*grad*-(DOPhOx)_12_ FRET-active NPs, were prepared by encapsulating both DiO (0.3 wt%) and DiI (0.3 wt%) into one NP. The concentrations of DiO and DiI were chosen to avoid the influence of self-quenching on the fluorescence signals. To confirm the occurrence of FRET, the fluorescence spectra of all four FRET-active POx NPs (800 µg/mL) were measured in PBS (pH 7.4) or acetone solutions at an excitation wavelength of 484 nm. A strong DiI signal (maximum fluorescence intensity at 565 nm) was observed for FRET-active NPs in PBS (red curves in Appendix A), which is due to the close proximity of DiO and DiI in the core of the NPs, resulting in the transfer of excited-state energy from DiO (donor) to DiI (acceptor). The FRET ratio, often used for quantitative analyses, is expressed as the ratio of the fluorescence intensity of the acceptor (I_A_) to the total fluorescence intensity of the donor and acceptor (I_D_ + I_A_) measured at their respective emission wavelengths using the excitation wavelength of the donor. The values of the FRET ratio in PBS for all four POx NPs ranged from 0.85 to 0.89, indicating efficient energy transfer. After dilution with an excess of PBS, the fluorescence emission maxima and FRET ratio remained unchanged for all four POx NPs (Appendix A). However, when diluted with an excess of acetone, the fluorescence intensity of all four POx NPs decreased sharply at 565 nm, whereas the fluorescence intensity increased at 501 nm (green curves in Appendix A), indicating the disaggregation of the NPs by acetone. FRET disappeared because DiO and DiI could no longer be tightly bound to each other, resulting in FRET ratios of 0.31, 0.27, 0.19, and 0.20 in (EtOx)_88_-*grad*-(MeOPhOx)_12_, (EtOx)_88_-*grad*-(EtOPhOx)_12_, (EtOx)_88_-*grad*-(HexOPhOx)_12_, and (EtOx)_88_-*grad*-(DOPhOx)_12_ FRET-active NPs, respectively. The disassembly of FRET active NPs in the presence of acetone was also confirmed by DLS measurements before and after solvent switching. 

Three key factors responsible for the disassembly of polymeric NPs after their systemic administration in vivo are rapid dilution, nanoparticle–unimer exchange dynamics, and the interaction with biological systems [55]. Previous studies have shown that the critical aggregation concentration (CAC) of a self-assembled nanostructure decreases significantly with the increasing length of the hydrophobic side-chain of gradient copoly(2-oxazoline)s [35] from 0.4 mg/mL to 0.015 mg/mL. Unloaded (EtOx)_88_-*grad*-(DOPhOx)_12_ NPs exhibited the lowest value of CAC. Polymeric NPs with a low CAC value should be more stable at rapid dilution in vivo and have a longer circulation time in the bloodstream. Hoogenboom and co-workers compared the micelle–unimer exchange dynamics of analogous amphiphilic block and gradient PMeOx-PPhOx [50] using FRET. The results showed that both types of copolymers exhibited dynamic unimer exchange, with the block copolymers manifesting higher stability than the gradient analogues.

Several studies based on FRET have shown structural and dynamic changes within polymeric NPs during their interaction with serum albumin (SA) [54]. Serum albumin (SA), the most abundant plasma protein, can affect the stability of nanoparticles and rapidly destroy their assembled structure. The strength of this interaction depends on the concentration of SA. To our knowledge, there have been no FRET-based studies to investigate the interaction of gradient copoly(2-oxazoline)s NPs with serum proteins and their effects on structural integrity. Here, we examined the interaction of four different FRET-active POx NPs with FBS (10%, *v*/*v*) in PBS mimicking the conditions of a cellular assay, HSA (35 mg/mL) in PBS, or γ-globulin (15 mg/mL) in PBS mimicking their concentrations in blood serum. The stability of POx NPs in PBS was measured as a control. The fluorescence spectra of four different FRET-active POx NPs were recorded at an excitation wavelength of 484 nm for 24 h after their incubation with FBS, HSA, or γ-globulin. The FRET ratio was calculated at each recorded time point and then normalized to time zero (beginning of recording). The largest changes in the FRET ratio were observed within 3 h of incubation (Appendix A). The decrease in the FRET ratio indicates the disassembly of POx NPs. In PBS, the dyes are not soluble and form some non-fluorescent aggregates. Therefore, the FRET signal is very weak unless they are encapsulated in the polymeric nanoparticles. Therefore, the disassembly of NPs can be monitored by measuring the FRET signal. All four types of POx NPs were stable for more than 20 h in PBS, whereas the POX NPs dissolved in 10% FBS solution, γ-globulin (15 mg/mL), and HSA (35 mg/mL) and the NPs with different lengths of hydrophobic side-chains showed different behaviors (Appendix A). Rapid disassembly of (EtOx)-*grad*-(MeOPhOx) NPs was observed within 30 min under all three conditions. Although a slight increase in side-chain length in (EtOx)-*grad*-(EtOPhOx) NPs prevented this rapid disassembly, the NPs lost approximately 50% of their FRET ratio within 3 h after incubation with FBS or HSA solutions. However, they showed strong resistance to γ-globulin solutions, as evidenced by minor changes in the FRET ratio. The behavior of (EtOx)-*grad*-(HexOPhOx) NPs was completely different: They showed very slow disintegration kinetics and very little change in FRET ratio even after 3 h of incubation in three solutions.

A further increase in the side-chain length of the (EtOx)-*grad*-(DOPhOx) NPs resulted in rapid disassembly, and the NPs lost approximately 25% and 35% of the FRET ratio within 2 h of incubation with HSA and FBS solutions, respectively. However, the NPs were relatively more stable in the γ-globulin solution and showed little change in the FRET ratio. The decrease in the FRET ratio of all four FRET POx NPs after 24 h incubation with PBS, HSA, FBS, and γ-globulin solutions is shown in Figure 3. From this, it can be seen that NPs were strongly disassembled in both cellular (10%, *v*/*v*, FBS in PBS) and blood-like concentrations of serum albumin (35 mg/mL). It has been previously reported that serum albumin is the main protein responsible for micellar disassembly and premature drug release [52,54,55]. Figure 3 shows that the length of the hydrophobic side-chain in the synthesized gradient copoly(2-oxazoline)s plays a crucial role in preventing the disassembly of NPs. When the side-chain length from ethyl to hexyl is increased, the strength of NPs to prevent disassembly is greatly improved. However, further increasing of the chain length from hexyl to dodecyl resulted in a decrease of the stability of the NPs. The thermodynamic stability of the self-assembled nanostructure followed the trend of chain length: methyl < ethyl < hexyl < dodecyl [35]. The present study shows how relatively small changes in the molecular structure can lead to substantial differences in the kinetic stability of POx NPs, with (EtOx)-*grad*-(HexOPhOx) NPs being the most stable followed by (EtOx)-*grad*-(DOPhOx), (EtOx)-*grad*-(EtOPhOx), and (EtOx)-*grad*-(MeOPhOx) NPs. Modification of the molecular structure of polyethylene glycol-based amphiphilic copolymers resulted in minimal interaction with albumin [59], and lipoprotein [61] NPs better retained their structure and payload in vitro and in vivo. Here, all four synthesized gradient copolymers contain EtOx as the hydrophilic part that forms the shell of NPs and interacts with proteins. Therefore, it is expected that they all have similar groups that interact with HSA. Therefore, the hydrophobicity of the core of POx NPs plays a decisive role in preventing their disassembly. The longer the hydrophobic side-chain of gradient copoly(2-oxazoline)s from methyl, ethyl, hexyl, and dodecyl, the greater the hydrophobicity of the core. (EtOx)-*grad*-(HexOPhOx) NPs were found to be an optimal system to achieve maximum curcumin loading with the highest kinetic stability. In the present study, disassembly experiments were performed on empty nanoparticles. Encapsulation of hydrophobic molecules such as curcumin further increased the hydrophobicity of the core of NPs. Therefore, it will be interesting to study such disassembly experiments with curcumin in the future.

### 3.3. Spectroscopic Properties of Curcumin-Loaded POx NPs

The absorption and emission spectra of curcumin are very sensitive to the change in polarity of the microenvironment [62]. Thus, they provide information about the hydrophobicity and heterogeneity of the core of polymeric NPs. To avoid the self-aggregation of curcumin, curcumin-loaded NPs with a curcumin concentration of 0.036 mg/mL were used for the spectroscopic measurements. The UV-Vis absorption and steady-state fluorescence emission spectra of the curcumin-loaded (EtOx)-*grad*-(MeOPhOx), (EtOx)-*grad*-(EtOPhOx), (EtOx)-*grad*-(HexOPhOx), and (EtOx)-*grad*-(HDOPhOx) NPs were measured in PBS (pH 7.4) (Appendix A). Various research groups have characterized the absorption and fluorescence emission spectra of curcumin in different solvents and media [62,63]. The absorption spectra of curcumin in PBS show a broad peak at 430–435 nm and a small shoulder at 355 nm, which can be attributed to the electronic π–π* transitions. During encapsulation in the non-polar cavity of POx NP, the shoulder at 355 nm disappeared and there is a larger peak at 430 nm. A new shoulder appeared at 455 nm, indicating that curcumin is in a completely different environment than in PBS. The value of the absorption maximum of curcumin encapsulated in NPs is shown in Table 3. 

When the length of the hydrophobic side-chain increased from methyl to dodecyl, the absorption maximum of curcumin is blue-shifted from 436 to 430 nm, indicating that curcumin is exposed to an increasingly nonpolar environment in the core of NPs. In an aqueous medium, the fluorescence quantum yield (Φ) of curcumin is very low (Φ = 0.001) [62] and the maximum value of Φ of curcumin is 0.2, confirming that non-radiative processes predominate over radiative processes. The Φ of curcumin in POx NPs was calculated using quinine sulphate in 0.1 N H_2_SO_4_ as a standard. Encapsulation in the NPs enhances the Φ, and the emission maximum is largely blue-shifted. When the length of the hydrophobic side-chain increased from methyl to dodecyl, Φ increased from 0.01 to 0.04 along with the 27 nm blue shift of the fluorescence emission maxima. Curcumin is in a less polar environment in the NPs than in the bulk water phase. The efficiency of the nonradiative deactivation process is largely reduced in the nonpolar cavity of POx NPs. Thus, curcumin inside (EtOx)-*grad*-(DOPhOx) NPs faces the least polar environment and inside (EtOx)-*grad*-(MeOPhOx) NPs faces the most polar environment.

### 3.4. Effect of Hydrophobic Side-Chain Length on the Cytotoxicity and Cellular Uptake of Curcumin-Loaded POx NPs 

The cytotoxicity of four unloaded or empty Pox NPs was investigated in our previous studies, which confirmed their bio-compatibility [35,36]. The cytotoxicity of curcumin-loaded POx NPs on U87 MG cells was detected by MTT assay. The MTT assay measures the metabolic activity of cells, which can be considered proportional to cell viability. Curcumin dissolved in DMSO added to the cell culture medium for 24 h resulted in a significant decrease in metabolic activity (60%) but at a concentration of 1 mM (Appendix A).

The cytotoxicity of NPs loaded with curcumin strongly depends on the length of the hydrophobic side-chain. Curcumin transported into cells with (EtOx)-*grad*-(MeOPhOx) and (EtOx)-*grad*-(EtOPhOx) NPs did not cause a significant decrease in the metabolic activity of cells in the available concentration range (limited by the method of the particle). In contrast, curcumin in (EtOx)-*grad*-(HexOPhOx) NPs caused the inhibition of metabolic activity to 60% at concentrations above 10^−5^ M (Appendix A). However, curcumin in (EtOx)-*grad*-(DOPhOx) NPs in the same concentration range inhibited the metabolic activity of cells to 60% after 24 h and 20% after 48 h (Appendix A). This is a much stronger effect than the application of curcumin alone.

The uptake of curcumin and curcumin loaded in NPs was detected by flow cytometry. Intracellular fluorescence of curcumin loaded into various POx NPs was detected in U87 MG cells 3 h after administration of NPs into U87 MG cells (Figure 4A). The most intense fluorescence was observed when curcumin was administered alone. However, curcumin in (EtOx)-*grad*-(DOPhOx) administered to the cells showed similar fluorescence intensities. Interestingly, the fluorescence of curcumin in the cells decreases with the duration of incubation (Figure 4B). Very low fluorescence was observed 24 h after curcumin or NPs administration. Low temperature (4 °C) often blocks active transport to cells. The uptake of curcumin by U87 MG was not temperature dependent, as shown in Appendix A. This suggests that curcumin enters cells by passive transport. This is also confirmed by the results of confocal fluorescence microscopy. 

The intracellular localization of curcumin in the endoplasmic reticulum was demonstrated by the Petit group [64]. In the present study, a homogeneous distribution of curcumin was observed in U87 MG cells after the administration of curcumin alone and encapsulated in (EtOx)-*grad*-(EtOPhOx), (EtOx)-*grad*-(HexOPhOx), and (EtOx)-*grad*-(DOPhOx) NPs (Figure 5). The localization of curcumin is very similar to the endoplasmic reticulum marker, as shown in Figure 5 (lower right image, labelled with fluorescent markers only in the absence of curcumin). A similar type of intracellular distribution of curcumin-loaded polymeric NPs was observed in the cytoplasm of HepG2 cells by Zhang et al. [65].

### 3.5. Pharmacokinetics of Curcumin-Loaded Pox NPs in Quail CAM

In recent years, the avian embryo chorioallantoic membrane model (CAM) has become a very popular and attractive in vivo alternative to the vertebrate model for the preclinical evaluations of various drug delivery systems [66,67,68,69,70]. Several studies have confirmed the advantages of the CAM assay such as viability, easy methodology, low cost, reliability, and reproducibility when compared with conventional mammalian models [66,67,68,69,70]. In our previous report, we investigated the biodistribution and phototoxicity of hypericin-loaded gradient POx NPs in CAM [35]. The increasing length of the hydrophobic side-chain in the gradient copolymers studied significantly affected the PDT efficacy and the release of hypericin from the nano-formulations. The fluorescence of curcumin can be used in photodiagnostics. In the present study, the pharmacokinetics of curcumin alone and loaded in the NPs were investigated in quail CAM. Intense fluorescence intensity of curcumin on the surface of CAM was detected immediately after administration (0 h) when curcumin was topically administered in (EtOx)-*grad*-(HexOPhOx) and (EtOx)-*grad*-(DOPhOx) NPs (see homogeneous yellow fluorescence within the application ring in Figure 6A).

The corona-shaped fluorescence maximum was observed at 5 h, as shown by the surface plots. The interaction of curcumin with the silicone ring (as a side-effect), which served as a carrier for the administration of the solutions, was observed throughout the observation period. The most intense curcumin fluorescence in the silicon ring was observed 24 h after administration. Traces of heterogeneously localized curcumin fluorescence on CAM (indicated by rings) were still observed after 5–24 h when (EtOx)-grad-(HexOPhOx) and (EtOx)-grad-(DOPhOx) NPs were administered. Curcumin administration did not significantly damage the tissues of CAM (see histology in Figure 6B). These results indicate that curcumin loaded in (EtOx)-grad-(HexOPhOx) and (EtOx)-grad-(DOPhOx) NPs was highly stable from the very beginning of administration.

## 4. Conclusions

In this work, we investigated the self-assembly and curcumin encapsulation ability of a series of amphiphilic gradient copoly(2-oxazoline)s, (EtOx)_88_-*grad*-(ROPhOx)_12_ with different hydrophobic side-chain lengths (R). As the side-chain length (R) increased from methyl to hexyl, the encapsulation efficiency of curcumin significantly increased. A maximum curcumin loading of 15.3 wt% was achieved with the (EtOx)_88_-*grad*-(HexOPhOx)_12_ copolymer at a curcumin concentration of 1 mg/mL in the feed. However, a further increase in the side-chain length of (EtOx)_88_-*grad*-(DOPhOx)_12_ resulted in a decrease in curcumin loading, likely due to crystallization induced by this larger alkyl group, which does not allow interaction with the drug. Thus, the length of the hydrophobic side-chain plays a crucial role in curcumin loading, and an optimal chain length of the alkyloxyphenyl (ROPhOx) group was required to achieve maximum drug loading. The kinetic stability of POx NPs in a biological medium monitored by FRET revealed the exquisite role of hydrophobic side-chain length. Human serum albumin (HSA) was found to be the key protein responsible for the rapid disassembly of (EtOx)-*grad*-(MeOPhOx) NPs. Meanwhile, (EtOx)-*grad*-(EtOPhOx) and (EtOx)-*grad*-(DOPhOx) NPs lost approximately 50% and 35% FRET efficiency, respectively, within 3 h after incubation with HSA. The best system was the (EtOx)-*grad*-(HexOPhOx) NPs, which maintained their structural integrity in the presence of HSA, FBS, and γ-globulin solutions. The spectroscopic properties of curcumin showed that it found the most hydrophobic environment inside the core of (EtOx)-*grad*-(DOPhOx) NPs followed by (EtOx)-*grad*-(HexOPhOx) NPs. 

Curcumin-loaded (EtOx)-*grad*-(MeOPhOx) NPs and (EtOx)-*grad*-(EtOPhOx) NPs showed no cytotoxicity to U87 MG cells, while the transport of curcumin with (EtOx)-*grad*-(HexOPhOx) NPs and (EtOx)-*grad*-(DOPhOx) NPs caused a significant decrease in the metabolic activity of cells, which was much greater than when curcumin was used alone. Cellular uptake in U87 MG cells and pharmacokinetics in CAM showed high curcumin fluorescence after the administration of (EtOx)-*grad*-(HexOPhOx) NPs and (EtOx)-*grad*-(DOPhOx) NPs compared with (EtOx)-*grad*-(MeOPhOx) NPs and (EtOx)-*grad*-(EtOPhOx) NPs. 

Due to the straightforward single-step preparation, gradient copoly(2-oxazline)s are gaining more and more interest as drug carriers compared to the analogous block copoly(2-oxazoline)s. The present study demonstrates how changes in the hydrophobic side-chain length in gradient copoly(2-oxazoline)s can lead to substantial differences in drug loading, stability, cytotoxicity, cellular uptake, and pharmacokinetics. We also propose that gradient copoly(2-oxazoline)s can potentially be considered effective next-generation polymeric nanostructures for biomedical applications.

## Data Availability

Data are contained within the manuscript.

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
