# Peer review of "Influence of Hydrophobic Side-Chain Length in Amphiphilic Gradient Copoly(2-oxazoline)s on the Therapeutics Loading, Stability, Cellular Uptake and Pharmacokinetics of Nano-Formulation with Curcumin"

_pharmaceutics, 2022, doi:10.3390/pharmaceutics14122576_

Round 1

Reviewer 1 Report

This comprehensive manuscript describing the use of amphiphilic gradient copoly(2-oxazoline)s to prepare a nanoformulation for curcumin is well written and the results presented seem valid. However, the paper represents another in a series of papers describing the use in formulations coating this polymer and the drug. This includes "Unravelling the Excellent Chemical Stability and Bioavailability of Solvent Responsive Curcumin-Loaded 2-Ethyl-2-oxazoline-grad-2-(4-dodecyloxyphenyl)-2-oxazoline Copolymer Nanoparticles for Drug Delivery" published in Biomacromolecules by the authors.  Reading this manuscript, it does not delineate the significance of the new results as it pertains to previously published studies. As such, the manuscript comes over as just another variation on the same theme. This is particularly troubling since the drug chosen is also not very novel. 

In addition:

There are numerous spelling and grammatical errors in the manuscript.

Author Response

Comment

This comprehensive manuscript describing the use of amphiphilic gradient copoly(2-oxazoline)s to prepare a nanoformulation for curcumin is well written and the results presented seem valid. However, the paper represents another in a series of papers describing the use in formulations coating this polymer and the drug. This includes "Unravelling the Excellent Chemical Stability and Bioavailability of Solvent Responsive Curcumin-Loaded 2-Ethyl-2-oxazoline-grad-2-(4-dodecyloxyphenyl)-2-oxazoline Copolymer Nanoparticles for Drug Delivery" published in Biomacromolecules by the authors.  Reading this manuscript, it does not delineate the significance of the new results as it pertains to previously published studies. As such, the manuscript comes over as just another variation on the same theme. This is particularly troubling since the drug chosen is also not very novel. 

In addition:

There are numerous spelling and grammatical errors in the manuscript.

 Reply

We are thankful to the reviewer for his/her valuable comments and suggestions to improve our manuscript.

We are agreed that the drug chosen is not very novel. However, very low solubility of curcumin in water (<8µg/mL) makes it an ideal model compound for our encapsulation study that aims to enhance the solubility of hydrophobic drugs.

In our previous work published in Biomacromolecules, 2018, only one type of gradient copolymer (with dodecyl side-chain) was used. The novelty of the present work is that gradient copolymers with varied side-chain length were used here for curcumin encapsulation and thereby investigated the influence of hydrophobic chain length on drug loading, stability and biological activity. Also the biodistribution of curcumin-loaded Pox NPs have been evaluated in the avian embryo chorioallantoic membrane (CAM) assay for the first time.

We have corrected spelling and grammatical errors in the revised manuscript. Please kindly see the revised manuscript.

Reviewer 2 Report

Datta et al. report on the properties of curcumin nanocarriers based on polyoxazoline amphiphilic gradient copolymers with EtOx as the hydrophilic segment and alkoxy phenyl oxazolines with varying length of side chains as the hydrophobic segment. The effects of chain length/hydrophobicity on curcumin encapsulation ability and loaded aggregate stability are elucidated by a gamut of techniques. Biocompatibility of nanocarriers, their stability in simulated physiological conditions and cellular uptake have been also studied and illustrated. The work is thorough and substantially original. The manuscript should be published after minor revisions as noted.

1. lines 49-50: in gradient copolymers the monomer distribution is varied along the main polymer chain.

2. Following results in Table 1 it would be informative to add the chemical structures of the copolymers studied in a separate scheme/figure.

3. Table 2: additional information needed regarding the presented samples. Please indicate CUR loading in each case. What are the sizes of empty copolymer micelles?

4. Lines 365-376: please discuss further reasons for these observations. Are there only crystallization or also side chain steric effects? Has crystallinity been confirmed within the aggregates? The results from the disassembly experiment on the copolymer aggregates should be taken also into account.

5. It seems that disassembly experiments were conducted on empty aggregates. What may be the effects of curcumin presence?

6. Fig. S1: intensity weighted size distributions should be also presented.

7. Some typos should be corrected.

Author Response

Datta et al. report on the properties of curcumin nanocarriers based on polyoxazoline amphiphilic gradient copolymers with EtOx as the hydrophilic segment and alkoxy phenyl oxazolines with varying length of side chains as the hydrophobic segment. The effects of chain length/hydrophobicity on curcumin encapsulation ability and loaded aggregate stability are elucidated by a gamut of techniques. Biocompatibility of nanocarriers, their stability in simulated physiological conditions and cellular uptake have been also studied and illustrated. The work is thorough and substantially original. The manuscript should be published after minor revisions as noted. 

Comment 1. lines 49-50: in gradient copolymers the monomer distribution is varied along the main polymer chain.

 Reply- We are thankful to the reviewer for his/her valuable comments and suggestions to improve the manuscript. We have corrected our statement in the revised manuscript. Please kindly see line numbers 49-52 in the revised manuscript.

Comment 2. Following results in Table 1 it would be informative to add the chemical structures of the copolymers studied in a separate scheme/figure.

Reply- We have added one new scheme, Scheme 1 presenting the chemical structures of all four gradient copolymers used for self-assembly and curcumin encapsulation.

Comment 3. Table 2: additional information needed regarding the presented samples. Please indicate CUR loading in each case. What are the sizes of empty copolymer micelles?

Reply- Additional information like solubilized aqueous curcumin concentration in each NPs has been added in the Table 2. The size of the empty copolymer nanoparticles was studied in our previous work and have been refereed here.

Comment 4. Lines 365-376: please discuss further reasons for these observations. Are there only crystallization or also side chain steric effects? Has crystallinity been confirmed within the aggregates? The results from the disassembly experiment on the copolymer aggregates should be taken also into account.

             Reply- Yes, we agree with the reviewer that such observations could be also due to the side chain steric effects. We have included this statement in the revised manuscript. However, the crystallinity has not been confirmed in the aggregates by any of the applied experimental method.

Comment 5. 5. It seems that disassembly experiments were conducted on empty aggregates. What may be the effects of curcumin presence?

       Reply- Yes, in the present study the disassembly experiments were performed on empty nanoparticles and we observed that the hydrophobicity of the core of the POx NPs played a decisive role to prevent its disassembly. Encapsulation of hydrophobic molecules like curcumin resulted in further increase in the hydrophobicity of the core of the NPs. Hence it will be interesting to explore such disassembly experiments with curcumin presence in future.

Comment 6. Fig. S1: intensity weighted size distributions should be also presented.

           Reply- Intensity weighted size distribution plot is now presented in Fig. S1b, supporting information.

Comment 7. Some typos should be corrected.

             Reply- Necessary corrections have been performed in the revised manuscript.

Reviewer 3 Report

The authors report on formation of polymeric drug carriers using amphiphilic gradient copolymers based on poly(oxazoline)s containing side alkyl chains of various length. They studied also delivery of encapsulated curcumin in such systems and dependance of performance of such drug carries on the length of the alkyl chains in a systematic way. It seems to be a reasonable extension of their works published in 2018 and 2021 (references 33, 34) where only one type of gradient copolymer (with dodecyl chains) was studied or other than curcumin active component was encapsulated.

The authors may also refer in the introduction to other amphiphilic polymer systems with grafted alkyl chains that were applied to form core-shell structures (nanoparticles, micells, capsules) and used as drug delivery systems. There are studies that refer also to the dependance of the size/stability of the formed structures on the chain lengths of pendant groups (e.g. J. Szafraniec et al, Nanoscale 2017, 9 , 18867-18880).

General comments:

It is hard to rationalized the average size equal to 176 nm for (EtOx)88-grad-(DOPhOx)12   (as reported in Table 2) from DLS result presented in Fig. S1 – the maxima of the presented curves for the systems with average size in the range 20-30 nm are not separated very far from the one assigned to the reported average of 176 nm. This issue should be clarified. It seems also that the applied filtration before the DLS measurements resulted in filtering off some larger structures, which are clearly visible in (cryo)-TEM images. It would not be possible to obtain an average diameter equal to 20-30 nm (see Table 2) if the structures up to 300 nm in length would be present in the dispersion. Thus, I would suggest repeating the DLS measurements but after filtration with larger filter (e.g. 0.45 um or larger pores) and/or no filtration.

Similarly, the curcumin loading capacity for the studied systems may have been affected by the applied filtering procedure. This issue should be clarified as some important conclusions are based on those results.

It seems that more important for the cytotoxicity studies is the type of applied polymer rather than the encapsulated curcumin. Thus, a proper control experiments with native polymer-based nanoparticles (without curcumin) should be presented and referred to.

Minor comments:

- Table 2. The results should be presented with proper accuracy, e.g. 21 +- 1 but not 1.06. It applies to both columns. It should be also specified in the caption what method was used to obtain those results.

- Figure S1. There are no spectra as it is written in the caption.

- Figure 3. Commas should be exchanged for dots in numbers.

Author Response

The authors report on formation of polymeric drug carriers using amphiphilic gradient copolymers based on poly(oxazoline)s containing side alkyl chains of various length. They studied also delivery of encapsulated curcumin in such systems and dependance of performance of such drug carries on the length of the alkyl chains in a systematic way. It seems to be a reasonable extension of their works published in 2018 and 2021 (references 33, 34) where only one type of gradient copolymer (with dodecyl chains) was studied or other than curcumin active component was encapsulated. The authors may also refer in the introduction to other amphiphilic polymer systems with grafted alkyl chains that were applied to form core-shell structures (nanoparticles, micells, capsules) and used as drug delivery systems. There are studies that refer also to the dependance of the size/stability of the formed structures on the chain lengths of pendant groups (e.g. J. Szafraniec et al, Nanoscale 2017, 9, 18867-18880).

Reply-We are thankful to the reviewer for his/her valuable comments and suggestions to improve the manuscript. We have referred other polymeric systems with grafted alkyl chains forming core-shell structures which was used as drug delivery systems (J. Szafraniec et al, Polymer 2020, 12, 1999). We have also referred J. Szafraniec et al, Nanoscale 2017, 9, 18867-18880 in the introduction which nicely demonstrated the dependence of the size/stability of the formed nanostructures on the chain lengths of pendant groups.

General comments: It is hard to rationalized the average size equal to 176 nm for (EtOx)88-grad-(DOPhOx)12 (as reported in Table 2) from DLS result presented in Fig. S1 – the maxima of the presented curves for the systems with average size in the range 20-30 nm are not separated very far from the one assigned to the reported average of 176 nm. This issue should be clarified. It seems also that the applied filtration before the DLS measurements resulted in filtering off some larger structures, which are clearly visible in (cryo)-TEM images. It would not be possible to obtain an average diameter equal to 20-30 nm (see Table 2) if the structures up to 300 nm in length would be present in the dispersion. Thus, I would suggest repeating the DLS measurements but after filtration with larger filter (e.g. 0.45 um or larger pores) and/or no filtration.

Reply- We have corrected the value of average size for curcumin-loaded (EtOx)88-grad-(DOPhOx)12 NPs in Table 2 which matches with the volume-weighted size distribution plot showed in Fig S1a, supporting information.

The intensity-weighted size distribution plot (Figure S1. b, Supporting Information) at 24 h showed the presence of larger NPs which constituted only a small volume fraction of the particles in the solutions. Kabanov and coworkers observed that poly(2-oxazoline)s-based block copolymer micelles could be elongated over time which is nicely supported by DLS and TEM measurements [ref 49]. In our case DLS result after 96 h (Figure S1. c, Supporting Information) suggested the increase in the population of elongated structures of curcumin-loaded NPs. TEM measurements were performed after 2 weeks from the time of NPs preparation. Thus, probably the concentration of worm-like NPs were increased during shelf-life of our sample. We have included this information in the revised manuscript.

Comment - Similarly, the curcumin loading capacity for the studied systems may have been affected by the applied filtering procedure. This issue should be clarified as some important conclusions are based on those results.

Reply -To calculate the curcumin loading capacity, immediately after dialysis the NPs solutions were filtered through both 0.22 and 0.45 μm syringe filter to remove the precipitated drug, which would provide false efficiency. The value of loading capacity was very close in both cases. Actually the large size particles were formed during shelf-life of our sample.

Comment -It seems that more important for the cytotoxicity studies is the type of applied polymer rather than the encapsulated curcumin. Thus, a proper control experiments with native polymer-based nanoparticles (without curcumin) should be presented and referred to.

Reply - The cytotoxicity of four unloaded or empty Pox NPs were studied in our previous study which confirmed their bio-compatibility [ref 35 and 36] and have been referred in the present work.   

Minor comments:

Comment- Table 2. The results should be presented with proper accuracy, e.g. 21 +- 1 but not 1.06. It applies to both columns. It should be also specified in the caption what method was used to obtain those results.

Reply – Necessary changes have been made in Table 2. The autocorrelation functions of the scattered intensity were analyzed by means of the Cumulants method to yield the Dh. This information has been added in the caption of Table 2.   

Comment- Figure S1. There are no spectra as it is written in the caption.

Reply – We have corrected the caption of Figure S1, supporting information.

Comment- Figure 3. Commas should be exchanged for dots in numbers.

Reply – Necessary changes have been made in Figure 3

Reviewer 4 Report

I think this is very well-written and well-structured manuscript with conclusions supported by experimentally obtained results. I enjoyed reading this work and recommend for publication as it is.

Author Response

Comment

I think this is very well-written and well-structured manuscript with conclusions supported by experimentally obtained results. I enjoyed reading this work and recommend for publication as it is.

Reply- Thank you very much for your comment and we are glad to know that you enjoyed reading our work.

Round 2

Reviewer 1 Report

The authors addressed the reviewer's concerns. In particular, the duplication of previously published papers.

Reviewer 3 Report

The authors correctly responded to the comments improving quality of the manuscript.